# Effectiveness of Interventions to Reduce Exposure to Parental Secondhand Smoke at Home among Children in China: A Systematic Review

**DOI:** 10.3390/ijerph16010107

**Published:** 2019-01-03

**Authors:** Yan Hua Zhou, Yim Wah Mak, Grace W. K. Ho

**Affiliations:** 1School of Nursing, Zhejiang Chinese Medical University, Hangzhou 310051, China; 15098644g@connect.polyu.hk; 2School of Nursing, The Hong Kong Polytechnic University, Hung Hom, Kowloon, Hong Kong 999077, China; grace.wk.ho@polyu.edu.hk

**Keywords:** secondhand smoke exposure, children, China, parental smoking, homes

## Abstract

There are health consequences to exposure to secondhand smoke (SHS). About two-thirds of children in China live with at least one person, usually a parent, who smokes at home. However, none of the reviews of interventions for reducing SHS have targeted children in China. The purpose of this study was to review the effectiveness of interventions for reducing parental SHS exposure at home among children in China. We searched various electronic databases for English and Chinese publications appearing between 1997 and 2017. Thirteen relevant studies were identified. Common strategies used in intervention groups were non-pharmacological approaches such as counseling plus self-help materials, and attempting to persuade fathers to quit smoking. Family interactions and follow-up sessions providing counseling or using text messages could be helpful to successful quitting. Several encouraging results were observed, including lower cotinine levels in children (*n* = 2), reduced tobacco consumption (*n* = 5), and increased quit rates (*n* = 6) among parents. However, the positive effects were not sustained 3~6 months after the interventions. Self-reported quitting without bio-chemical validation was the most common outcome measure. A study design using biochemical validations, a longer follow-up period, and targeting all people living with children in the same household is recommended.

## 1. Introduction

Second-hand smoke (SHS), also called passive smoke, or environmental tobacco smoke, is defined as “the combination of smoke emitted from the burning end of a cigarette or other tobacco products and smoke exhaled by the smoker” [1]. A growing body of evidence suggests that exposure to SHS has negative consequences for both children and society. For example, children exposed to SHS are more likely to develop middle ear disease, respiratory symptoms, impaired lung function, lower respiratory illness, and sudden infant death syndrome [2]. A child’s exposure to SHS at home during the primary school years is also a predictor of smoking initiation [3]. At the societal level, exposure to SHS places a heavy economic burden on the healthcare system. SHS exposure-related health conditions contribute significantly to healthcare expenditures for children and adults. For example, medical and other indirect costs of SHS exposure among children living in public housing were estimated to be approximately $182 million in the US in 2011, while the corresponding figure for adult public housing residents was over $123 billion [4].

China has the world’s largest population of tobacco consumers. In 2009, smokers in China consumed nearly 40% of cigarettes in the world, more than the next top four tobacco-consuming countries combined [5]. Furthermore, the rate of smoking cessation is extremely low in China. An international survey conducted from 2008 to 2010 showed that the successful quit rate among Chinese smokers was 12.6%, which was far lower than the figures for developed countries (e.g., 57.1% in the UK and 48.7% in the US) and developing Asian countries (e.g., 20.9% in the Philippines and 28.4% in Thailand) [6].

China ratified the Framework Convention on Tobacco Control (FCTC) in 2005, and the prevalence of SHS exposure has been significantly reduced since the implementation of tobacco control strategies such as smoking bans in public areas (e.g., cultural sites and commercial places); however, non-smokers remain unprotected from exposure to SHS at home [7]. In fact, people may be smoking more at home now that smoking in public areas is banned [8]. In China, more than 100,000 people die each year as a result of exposure to SHS [9]. Compared with adults, children are more likely to be exposed to SHS within the home. This may be because children spend most of their time at home, and are usually unable to impact the smoking behaviors of family members. For example, Wipfli and colleagues [10] recruited mother-child dyads from 31 countries and found a higher concentration of air nicotine in homes with smokers; among the exposed mother-child dyads, higher concentrations of nicotine were detected in the hair of the children than in the mothers.

A recent international study of over one billion children showed that 66.7% of children in China younger than 15 lived with someone who smoked inside their home, which was the highest prevalence of child SHS exposure in the home in the 21 countries that were included in the survey [11]. Parental smoking status and smoking patterns at home are the key factors in the SHS exposure of children [12]. There is no safe level of exposure to SHS [13]. Therefore, understanding the effectiveness of interventions to reduce children’s SHS exposure, particularly at home, might be helpful in developing protocols for creating a smoke-free environment for children in China. Effective interventions for reducing children’s exposure to SHS at home could greatly enhance the health of parents and children, reduce morbidity, and relieve the burden on the healthcare system.

Several systematic reviews have been conducted to summarize the effectiveness of interventions to reduce SHS exposure among children of different age groups, including those younger than 12 months [14], under 5 years [15], 0–6 years [16], 0–12 years [17,18], and under 18 years old [19]. Five of the six reviews targeted studies of parents or primary caregivers [14,15,16,18,19]; another included studies that targeted healthcare providers and teachers [17]. However, none specifically targeted Chinese families. It is possible that interventions that are effective in Western settings may not be practicable in the Chinese context due to differences in culture and smoking behaviors [20].

## 2. Objective of the Present Study

The purpose of this study is to systematically review the evidence on the effectiveness of all types of interventions for reducing exposure to SHS at home for children living with their smoking parents in China. The findings will act as a reference for designing and assessing the feasibility of strategies to reduce the exposure of Chinese children to parental SHS at home.

## 3. Methods

The PRISMA Statement for Reporting Systematic Reviews and Meta-Analyses was employed to guide this review.

### 3.1. Search Strategy

A systematic search of the literature was carried out to identify relevant studies published in English between January 1997 and December 2017. The following databases were used: PubMed, MEDLINE (via Ovid), CINAHL Plus with Full Text (via EBSCOhost), EMBase (via Ovid), and the Cochrane Central Register of Controlled Trials (CENTRAL). The keywords for our search included a combination of terms related to secondhand smoking exposure (i.e., secondhand smoke, environmental tobacco smoke pollution, passive smoke, involuntary smoke, third-hand smoke exposure, and tobacco smoke pollution) and terms related to Chinese children and their families (i.e., child, infant, baby, newborn, home, smoke-free home, family, household, smoke-free household, China, Chinese, Hong Kong, Taiwan, Macau). To retrieve relevant studies published in the Chinese literature, the same key words in Chinese were used when searching the China National Knowledge Infrastructure (CNKI) database and Wanfang MED Online. We searched the Chinese publications appearing in Wanfang MED Online from the conception of the database in 1998 to 2017. The reference lists of the selected articles were subjected to a hand search to identify additional articles.

### 3.2. Selection Criteria

Studies were eligible for inclusion if they meet the Population, Intervention, Comparison, Outcome, and Study design (PICOS) criteria, namely: (1) population: Chinese parents who smoke at home and who have a child aged 0 to 18 years living with them in the same household in China; (2) intervention: all types of interventions that aimed at helping parents reduce their children’s exposure to tobacco smoke at home, and with reports on the intensity of the intervention and the length of the follow-up; (3) comparison: studies with two or more comparison groups (i.e., the usual care, the intervention and/or an alternative intervention), or one group pre-post comparison; (4) outcome: reduction of SHS exposure among the children at home or changes in the smoker’s smoking behaviors at home; and (5) study design: interventional study. For publications reporting on the same trial, we only included the one that contained the most information that would contribute to addressing the objectives of the review; the other publications were still reviewed to help with understanding the trial in more detail. Studies were excluded if they were: (1) not published in English or Chinese; or (2) not available in full text.

### 3.3. Study Selection

Two investigators independently assessed and identified relevant articles to be included in this review. Studies were screened by title and abstract. Irrelevant or duplicate articles were excluded, and all remaining articles were subjected to full-text screening. A librarian also lent assistance in conducting a thorough search for the relevant articles. Differences between the reviewers in the inclusion of articles were resolved through discussion and consensus. Figure 1 depicts the process of screening and including articles, and lists the reasons for excluding articles.

### 3.4. Data Extraction

One author extracted the following information from the included studies: (1) authors and years of publication; (2) city/area where the study was conducted; (3) study design; (4) participant characteristics (target population, age and health conditions of the children, number of participants, recruitment setting); (5) intervention details (intervention components, mode of delivery, intervention intensity, total contact time, provider, length of follow-up, and theoretical framework); (6) comparison: intervention and control group, if applicable; (7) outcome measures (self-reported and biochemically validated outcome measures); and (8) study quality. Another author checked the extracted data.

### 3.5. Study Quality

Tools developed by the American National Heart, Lung, and Blood Institute were applied to assess the quality of the reviewed studies [21]. Separate tools were used to assess the quality of different study designs. Each tool included 12–14 questions to assess bias, confounding, power, and the strength of the association between the intervention and the outcomes. Study quality was divided into three levels: good, fair, and poor. One reviewer independently assessed the quality of all of the included studies, and another checked the results for accuracy.

## 4. Results

### 4.1. Study Characteristics (n = 13)

A final sample of 13 studies was included in this review (Appendix A). These studies were conducted in nine places: Beijing [22,23], Changchun [24], Fenyang [25], Guangzhou [26,27], Hong Kong [28,29,30], Nanning [31], Shanghai [32], Taiwan [33], and Tianjin [34]. Study designs include randomized controlled trials (*n* = 7) [24,28,29,30,32,33,34]; control trials without randomization (*n* = 2) [22,25]; and pre-post studies (*n* = 4) [23,26,27,31]. The target subjects of the interventions included household members (*n* = 6) [22,23,25,26,31,32]; family units (i.e., non-smoking mothers and smoking fathers) (*n* = 3) [24,29,30]; smoking parents (*n* = 2) [28,34]; and parent-child dyads (*n* = 2) [22,27]. The study participants were recruited from maternal-child health centers [22,23,24,28,30], primary schools [26,27,33], community settings [24,25,35], or hospitals [29,31].

#### 4.1.1. Intervention Arms and Components

Eight studies [22,25,28,29,30,32,33,34] had one intervention group and one comparison group; while one study had two intervention groups: a group with a counseling intervention and a group with a counseling plus text messages intervention [24]. Four studies were one-group pre-post studies [23,26,27,31].

The interventions involved the following components: (1) self-help materials (*n* = 11) [22,23,24,27,28,29,30,31,32,33,34], which included information about the health risks of SHS exposure, instructions for creating a smoke-free home, protecting children from SHS exposure, and tips for smoking cessation; (2) prompts to warn against smoking at home (e.g., a poster that says “Dad, the smell of your smoke makes me uncomfortable” or a card emphasizing that smoking is forbidden in any place inside the house) (*n* = 6) [24,29,30,31,32,33]; (3) individual counseling (*n* = 6) [28,30,31,32,33,34]; (4) text messages via phone (*n* = 1) [24]; (5) group counseling (*n* = 3) [24,30,33]; (6); biochemical feedback (*n* = 1) [33]; (7); and health education (*n* = 5) [22,25,26,27,29].

#### 4.1.2. Mode of Delivering the Intervention and Providers of the Intervention

More than half of the studies used a single mode to deliver the intervention, including telephone-based counseling (*n* = 1) [31], face-to-face health education (*n* = 3) [23,25,26], face-to-face health education and hand-delivered materials (*n* = 2) [22,27], and face-to-face counseling and hand-delivered materials (*n* = 1) [34]. Among the five studies that used more than one delivery approach, three used a face-to-face and telephone-mediated intervention [29,30], one used telephone counseling and mailed materials [28], and one used face-to-face and telephone counseling with hand-delivered/mailed materials [32].

Eleven out of the 13 studies reported the providers of the intervention. Trained community healthcare workers were the interventionists in four studies [22,23,24,32]. The two studies conducted in Hong Kong employed trained nurses to deliver the intervention [28,30]. In one study of parents with sick children, the intervention was delivered by nurses from pediatric wards or the outpatient department [29]. Other intervention providers included an instructor specialized in psychiatric nursing and experienced in clinical instruction, group therapy, and psycho-education [33], a master’s student [34], trained junior pediatricians [31], and trained teachers from the target primary schools or trained workers from the school infirmary [27].

#### 4.1.3. Intervention Intensity

The number of sessions in the interventions ranged between one to six sessions. Four interventions consisted of one to two sessions [24,26,27,29] and nine interventions delivered three or more sessions [22,23,25,28,30,31,32,33,34]. Seven studies [23,27,28,29,30,31,32] reported the total contact time, which ranged from 9 to 15 min [23] to approximately 240 min [30].

#### 4.1.4. Length of Follow-Up

All of the studies followed the participants over time after the completion of the intervention, with the exception of one study that evaluated the immediate post-intervention effects [23]. The length of the follow-up was defined from the initial contact to the time of the final follow-up assessment. Eight studies evaluated the effectiveness once, and the follow-up period was 21 months [26], 6 months [22,28,32,34], 3 months [31], or 1 month [27]. Five studies measured their outcomes twice, and the follow-up interval was 6 months and 12 months [24,25,30], 3 months and 12 months [29], and 8 weeks and 20 weeks, with the exception of a final assessment of the children’s urine, which was carried out at 6 months because of the unavailability of equipment [33].

#### 4.1.5. Theoretical Framework

In five studies, the interventions were developed based on the following theoretical frameworks: the protection motivation theory [32]; the trans-theoretical model [28]; the theory of planned behavior [29]; and the Five A’s approach (Ask, Advise, Assess, Assist, and Arrange) [31,34]. Two studies included more than one theoretical underpinning: the transtheoretical model and the I-change model [33]; and the transtheoretical model, social cognitive theory, and social ecological theory [30].

#### 4.1.6. Control Group Intervention

In eight of the nine studies involving comparison group(s), the participants in the control group received the usual care without a tobacco control intervention or cessation services [24,29], with very brief advice on the hazards of tobacco and quitting smoking [22,30], printed self-help materials [32] or only printed materials on the hazards of tobacco [31], the benefits of quitting smoking [28], or child nutrition issues [34]. The remaining study did not report the intervention for the control group [25].

### 4.2. Intervention Outcomes Related to Tobacco Consumption by Parents

#### 4.2.1. Quit Rate

Five studies measured the quit rate based on self-reports. A significantly higher self-reported quit rate among the parents in the intervention groups was observed in three studies [22,24,34]. In the pre-post study by Huang and colleagues [31], six out of 87 (7%) participants reported that they had quit smoking at the 3-month follow-up. In another study that measured the quit rate as reported by non-smoking mothers, a significantly higher quit rate in intervention group was detected at 3 months, but a significant group difference were not observed at 12 months [29].

Three studies also employed bio-chemical tests to confirm self-reported quitting. In the study by Abdullah and his team in Hong Kong, a significantly higher quit rate was reported by the smokers and confirmed by the results of a biochemical validation test [28], while no group difference in both the self-reported and biochemically validated quit rate was reported in another study by Abdullah in Shanghai [32]. Yau [30] found a contradiction between the self-reported quit rate and the biochemical results. In her study, significantly more smokers in the intervention group (82 out of 598) than in the control group (45 out of 560) reported that they had not smoked during the previous 7 days at the 12-month follow-up. Although the results of the biochemical tests of all 34 fathers who participated in the tests were found to be consistent with their self-reported results, no group difference was detected in the biochemically validated quit rate (i.e., 3.5% for the intervention group and 2.3% for the control group).

#### 4.2.2. Other Smoking-Related Results

Some other smoking-related outcomes were reported. For example, a significant reduction in total tobacco consumption or tobacco smoked at home was observed in several control trials [28,29,30,32] and in one pre-post study [23] immediately after the intervention. However, two of the studies that included a longitudinal follow-up found that the group difference in tobacco consumption was not sustained at 12 months [29,30]. Separately, four studies reported their intervention had no effect on reducing tobacco consumption [22,25,26,27]. Besides that, some findings demonstrated that significantly more of those in the intervention group indicated a forward progression in stage of readiness to quit smoking at the 6-month follow-up [28,30], although the effect was not maintained at the 12-month follow-up [30], and made attempts to quit for at least 24 h at 3 months [29] or 6 months [28]. In another study in Hong Kong, more participants who had joined the family counseling session made attempts to quit than did the other smokers in intervention group and those in control group, while no difference was found among those who had not received the family counseling session, regardless of the group allocation [35].

### 4.3. Intervention Outcomes Related to Child and Family Health

#### 4.3.1. Children’s Health and Behaviors

Positive effects on the health and behaviors of the children were measured in four studies based on parent- or child-reported outcomes. Compared with the control group, the parents in the intervention group reported that their children had significantly fewer respiratory symptoms [32] and had been admitted less frequently to hospital [34] at the 6-month follow-up. Significantly more children in the intervention group reported at the 20-week follow-up that they had frequently used strategies to avoid SHS exposure [33]. In a pre-post study by Liu et al. [26], the rates of children’s exposure to SHS decreased from 56.7% at baseline to 51.7% at the 21-month follow-up.

However, inconsistencies were found between three studies that used biochemical validation to evaluate the children’s SHS exposure. In the studies by Chen et al. [33] and Abdullah et al. [32], a significantly lower urine cotinine level was detected in children in the intervention group than in those in the control group at the 6-month follow-up. Meanwhile, Yau [30] reported no significant group difference in the saliva cotinine of infants at both the 6-month and 12-month follow-up, although a significant decrease was observed in both groups across time. The age of the recruited children (0~18 months) and the intervention offered to the control group (including verbal advice on the harms of SHS and tips for quitting smoking) in that study might have weakened the difference between the intervention and control groups.

#### 4.3.2. Parent Practiced not Smoking at Home

Three studies reported that significantly more families in the intervention group [24,28,32] or after the intervention [27,31] adopted a no-smoking policy at home. In the study in Taiwan, both the parents and children reported a decrease in parental smoking at home at two follow-ups, but there was no significant difference in the effect when the group difference across time was taken into account [33]. In the study by Yau and her team, no between-group difference was observed between those families who had adopted a policy of completely restricting smoking and those with fathers who smoked at home [30], but significantly fewer fathers who had received a family counseling intervention smoked at home than those who had not, regardless of which group they had been allocated to [35].

Changes after adopting a partial smoking ban at home were seen in several studies. Three studies featuring randomized controlled trials found that there were more household members who did not smoke around children among the participants of the intervention group than among those of the control group [28,32,33]. In the pre-post study conducted in Beijing, the number of participants who did not smoke in the presence of children increased from 69% to 85% [23]. However, no significant change or group difference was found in the two studies [22,27].

Some studies surveyed parental strategies for protecting their children from SHS exposure before and after the intervention. In the study in Taiwan, the percentage of parents who engaged in high-frequency preventative behavior increased in the intervention group by 10.86% from baseline, while decreasing by 10.52% in the control group [33]. Two studies found that, following the intervention, more parents took action to prevent their children from SHS exposure at home, such as by opening a window [27] and not allowing others to smoke around their children [31].

### 4.4. Quality Appraisal

Our review included four studies that conducted an intention-to-treatment (ITT) analysis of several outcome measures, including the quit rate [28,29,30], the reduction in daily cigarette consumption [28,29], quit attempts [28,29,30], stages of readiness to quit [28,30], restrictions on smoking at home [28], smoking behavior at home [33], and the use of prevention strategies by parents and children [33]; and four studies that applied biochemical tests to confirm self-reports of quitting [28,30,32,33] (Table 1 and Table 2). Five studies were rated to be of fair quality: The dropout rate in both study groups was over 20% in the study by Yau and her team; also unreported was whether the intervention providers knew about the group assignments [30]. In the study in Taiwan, the authors did not report that the sample size was sufficient for their study objectives, and the blinding of the participants and interveners was also not reported [33]. In the study by Abdullah et al. in Hong Kong, the method for calculating the sample size was not reported [28]. Outcome measures were not carried out multiple times after the interventions in two pre-post studies [27,31].

Eight studies were considered to be of poor quality for the following reasons: (1) the dropout rate in both study groups was over 40%, and no ITT analysis was carried out (*n* = 1) [32]; (2) participants who were using other services or forms of support for smoking cessation were not excluded from the study (*n* = 1) [24]; (3) all outcomes relating to the smoking behavior of a child’s father were reported by non-smoking mothers (*n* = 1) [29]; (4) group differences in demographic characteristics and smoking-related characteristics at baseline were not reported (*n* = 1) [34]; (5) some important details of methodology such as the method of randomization and/or the criteria for the selection of the study population were not reported, which may lead to doubts about the reliability of the results (*n* = 4) [22,23,25,26].

## 5. Discussion

### 5.1. Principal Findings

We found a total of 13 trials carried out in three Chinese jurisdictions: Taiwan, Hong Kong, and Mainland China. Our findings show that all of the studies demonstrated positive effects on promoting a smoke-free home to some degree. However, both the number and the quality of the studies remain limited. First, we found that few intervention studies on reducing in-home SHS exposure among children in China have been carried out, and that most of them were conducted in urban cities and none in rural areas. Second, most studies evaluated the efficacy of their intervention based on self-reported outcomes, which were observed to be inconsistent with biochemical confirmations.

### 5.2. Methodology

#### 5.2.1. Intervention Strategies and Approaches

The most common intervention strategy involved a combination of self-help materials and a counseling service, which was commonly provided by community health workers and trained nurses. Our findings demonstrated that most trials using this approach led to significantly higher rates of short-term abstinence among the smoking parent than other approaches. Smokers with a history of 24 h of abstinence in the past year and those who have tried more methods of stopping smoking are more likely to achieve a longer quit period in smoking cessation programs [36]. Given that only 22.4% of pediatricians in China regularly advise parents to quit smoking and that only 3.8% of pediatricians arrange a follow-up appointment to assess the effects of their advice [37], it might be practical for community health workers and nurses in China to provide smoking cessation services. However, a lack of training in reducing tobacco consumption and in cessation counseling is a primary barrier to the provision of SHS exposure reduction services by pediatric nurses [38]. A large scale survey conducted in Hong Kong (*n* = 4413) revealed that the majority of nurses recognized that tobacco control is an importance health advocacy program, but only 40.6% of the nurses believed they are equipped with the knowledge and skills to help smokers to quit smoking and about 13.5–64.9% of the nurses would practice the recommended 5As (Ask, Advise, Assess, Assist and Arrange) to help smokers to quit smoking [39]. In Shanghai, only 25% of community general practitioners have ever undergone training in smoking cessation interventions [40]. Therefore, providing such smoking cessation training to community workers and nurses and establishing a standard of care for them might promote the practice of smoking cessation counseling in China.

None of the studies used smoking cessation medications in their interventions, but prior studies have shown that integrating counseling with drug therapy may be more effective for smoking cessation than using counseling alone. For example, Grassi et al. [41] allocated participants into two groups, namely, an intervention group that received six weeks of group counseling and a daily administration of bupropion for seven weeks and a control group that received counseling alone. The smoking abstinence rate of smokers in the intervention group who had and had not completed all of the drug therapy was 68% and 56.6%, respectively, while that of the participants in the control group was 35.3%. Positive outcomes were also reported in China [42]. Although smoking cessation drugs such as bupropion [43] and varenicline [44] have been shown to be effective and safe in Chinese populations, there is a persistently low uptake of smoking cessation drugs, which is perhaps attributable to a deep-rooted cultural belief in China emphasizing that every medicine has its adverse effects [45]. Financial pressure is another possible explanation, because the cost of smoking cessation drugs is not covered by medical insurance in China, which might make the drugs unaffordable for smokers. It is possible that strengthening the knowledge of smokers on pharmacological treatments for smoking cessation may increase their use of smoking cessation medications to support quitting. Alternatively, interventions involving traditional Chinese medicine techniques, such as Auricular-Point-Pressing Therapy, should also be explored, as they have been shown to increase tobacco abstinence rates [42].

In recent years, promising outcomes, such as weight loss, have been observed from the use of smartphone applications by health professionals in China to promote healthy lifestyles to patients [46]. A recent review showed that the intervention group using a short messaging service achieved a 35% higher quit rate than the control group [47]. Therefore, an mHealth intervention approach might be a feasible and effective for delivering interventions to parents in China, especially to parents living in rural areas, who usually have limited access to smoking cessation services.

Quitting smoking might not be the only solution to decreasing the exposure of children to SHS at home. We identified one intervention in Taiwan [33] that was aimed at improving the protective strategies used by smoking parents rather than at getting them to quit smoking, which resulted in a significant decrease in the SHS exposure of children. These positive results were observed in another study by Hovell et al. in California [48]. Given the difficulties of quitting smoking, these findings suggest that interventions that are designed to enhance protective strategies, rather than at achieving complete smoking cessation, warrant further testing in China, especially among parents who are resistant to the idea of quitting smoking or who cannot stop smoking within a short period of time.

#### 5.2.2. Intervention Target

All of the reviewed studies, only one smoker was recruited from in each family except the one conducted in Hong Kong [28], regardless of the number of smokers in the household, which might partly explain the discordance between the participants’ smoking status and the children’s cotinine levels. Studies have found that a child’s level of exposure to secondhand smoke exposure can be affected by the number of smokers living in the home with the child [12,49]. Therefore, interventions that target all smoking members living with children are warranted in the future, especially in a Chinese society where grandparents commonly live in the same household and play an important role in taking care of their grandchildren.

Encouraging interactions between smokers and their children or spouse during the smoking cessation period could facilitate quitting. Two studies included smoking fathers and their children or wives in their intervention programs [24,33], and both studies showed better outcomes for the counseled groups than for the controls. Another study encouraged the couples in the intervention groups to participate in a family counseling session [30]. Those who had taken part in the counseling session reported better outcomes than those who had not (i.e., a higher self-reported 7-day point prevalence quit rate, less smoking at home, more attempts at quitting, fewer cigarettes consumed per day, and more help and psychological support offered by mothers to fathers) [35].

All of the identified studies targeted smoking fathers to reduce the secondhand smoke exposure of their children. For a Chinese man, identifying as a male protector could greatly promote the voluntary cessation of smoking when the man becomes a father [50]. However, in traditional Chinese cultural norms, smoking is regarded as a symbol of a man’s personal freedom and social position [51], and Chinese women still hold a tolerant attitude towards men who smoke at home due to concerns about family harmony [52]. A study in China observed a significant difference in biochemically verified smoking abstinence between smokers with and without a family member who had received guidance in helping them to quit [53]. Significantly better results were also reported in the daily smoking of tobacco, quit attempts, and communication between smokers and their family members in the intervention group. Therefore, smoking cessation programs are required, including interventions to promote knowledge and strategies to non-smoking mothers on supporting smoking cessation and establishing a smoke-free home.

#### 5.2.3. Intervention Effects in the Long Term

Most interventions achieved short-term success in changing the perceptions and behaviors of smokers and non-smokers, but most studies reported that the effects of the intervention were not sustained in the long run. The smoking relapse rate was high. Different from the situation in Western countries, where relapses are commonly triggered by personal feelings such as stress and cravings [54], the important factors leading to relapse among Chinese smokers are being in social settings involving other smokers and peer influence [55]. In China, it is also good manners to share cigarettes with others, even strangers, and to give cigarettes as gifts (e.g., as a demonstration of filial respect and love for elders) [56]. These findings suggest that more research is needed to understand whether and how wider public health initiatives aimed at changing attitudes towards smoking, as well as the implementation of tobacco-control policies that discourage the giving of cigarettes as gifts, may help to sustain the effects of smoking cessation interventions in the long-run. Other relapse-prevention strategies should also be developed to sustain the effects of smoking cessation interventions. These include self-help materials, telephone follow-ups over a long period [57], and smoking cessation medications [58].

#### 5.2.4. Methods and Measurements

More than half of the included studies used a follow-up interval of 6 months or less to evaluate the effectiveness of their intervention. However, previous studies have shown that 6 months is not a good proxy for smoking abstinence in the long run and that many smokers who successfully quit relapse after one year of abstinence [59,60]. Although a reliable point for life-time abstinence has not been identified, Nohlert et al. [61] reported that abstinence from smoking by the 12-month follow-up could significantly predict people’s smoking status in the long term, while Ockene et al. suggested that a follow-up period of one or two years would be more appropriate [60].

More studies assessing biochemically confirmed outcomes in terms of SHS exposure are also needed in the future, since self-reported measures often underestimate the actual situation of SHS exposure when compared with biochemical verification [62]. However, it should be noted that heavy SHS exposure outside the home environment can also interfere with the results of a biochemical validation. For example, a study showed that children who had spent most of their time in the hospital on the day before a urine sample was collected had a significantly lower level of urine cotinine than those in other settings [49]. Moreover, the characteristics of children should be taken into account when selecting the cotinine cut-off point of SHS exposure in children, because the rate at which nicotine is metabolized can vary by race (e.g., it is faster in Caucasians than in Asians) [49] and age (i.e., it is faster in youth than in toddlers) [49].

### 5.3. Strengths and Limitations of the Study

To the best of our knowledge, this is the first systematic review to describe and summarize the effectiveness of interventions to reduce SHS exposure at home for children in China. The findings of the present review provide a deeper understanding of the types of smoking cessation interventions or strategies that are effective at reducing the exposure of children to tobacco smoked by their parents in a country where the prevalence of smoking is high, the quit rate is low, and the majority of children are exposed to SHS at home. We also assessed the strengths of the identified studies in order to make recommendations for future research. Our study has several limitations. First, although we made attempts to find additional papers through hand searches, it is possible that other interventional studies might have been conducted but never published. Second, we only included full-text articles published from 1997 to 2017. It is possible that other studies that were not published during this period were not identified, but no study was excluded from this review because the full text of it was unavailable. Third, most of the studies that were included in this review were conducted in well-developed cities in China such as Hong Kong and Shanghai. More studies from developing cities could be helpful at addressing SHS exposure among children in China, particularly as tobacco use is more popular in those areas. Fourth, the limitations in the methodology employed in several selected studies, such as those concerning the inclusion criteria and the characteristics of the study population at baseline, were not described, which might affect the generalization of the findings of this review. Finally, we were unable to carry out a meta-analysis due to the heterogeneity of the data between different articles.

### 5.4. Comparison with Other Studies

Different from the reviews in Western countries, where most interventions were designed to reduce SHS exposure from mothers who smoke [14,15,17], the studies included in this review were more likely to target fathers who smoke. This may due to differences in smoking prevalence between males and females in China (52.9% vs. 2.4%); whereas the corresponding ratios are similar in Western countries [63]. Similar to the findings in our study, the interventions that were examined in previous reviews ranged widely in content. They also included strategies such as the use of medications [16,18,19] and air cleaners [16,18], which were not found in our study. Self-reported outcomes were the most commonly used measures of outcome, both in our study and in previous reviews, but only one review included studies that reported levels of air nicotine or particulate matters [18].

## 6. Conclusions 

The present review provides the first comprehensive examination of the effectiveness of interventions for protecting children from SHS exposure at home in China. All of the interventions in these studies demonstrated a positive outcome, such as short-term abstinence from smoking and the implementation of a complete no-smoking policy at homes. However, they did not demonstrate positive or sustained effects in terms of an increase in quit rates or a decrease in the biochemical markers of SHS exposure among children. Further trials introducing other approaches, such as the addition of smoking cessation drugs or other protective strategies, and targeting all of the smokers in a home, are needed in the future. Also recommended is a study design that employs a longer follow-up period and biochemical techniques.

## Figures and Tables

**Figure 1 ijerph-16-00107-f001:**
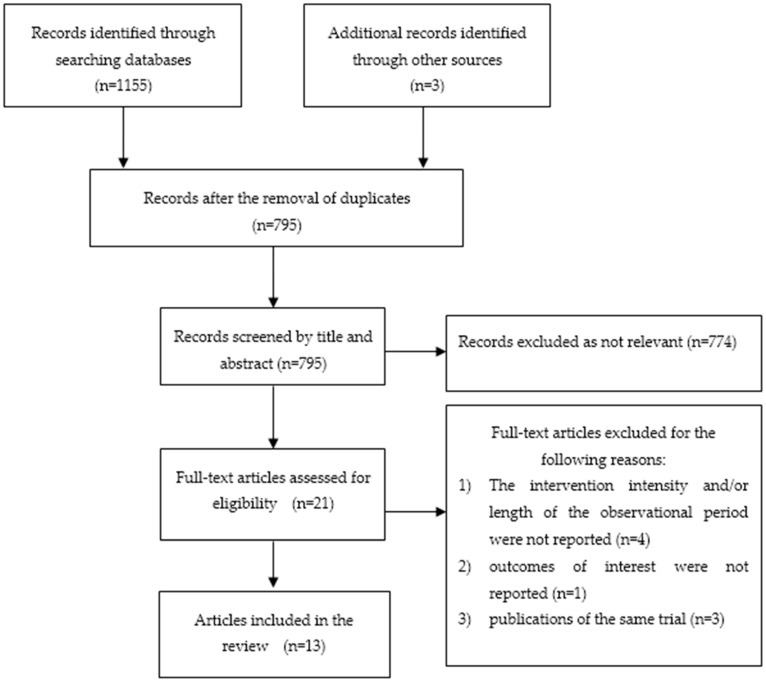
Flow Diagram of Included and Excluded Studies.

**Table 1 ijerph-16-00107-t001:** Quality Assessments of Studies Included in the Systematic Review (controlled intervention studies).

	Abdullah et al., 2015 [32]	Yu et al., 2017 [24]	Yau, 2011 [30]	Chen et al., 2016 [33]	Abdullah et al., 2005 [28]	Chan et al., 2008 [29]	Yang, 2007 [34]	Li, 2000 [22]	Liang & Wang, 2001 [25]
1. Was the study described as randomized, a randomized trial, a randomized clinical trial, or an RCT?	Yes	Yes	Yes	Yes	Yes	Yes	Yes	No	No
2. Was the method of randomization adequate (i.e., use of randomly generated assignment)?	Yes	Yes	Yes	Yes	Yes	Yes	Yes	NR	NR
3. Was the treatment allocation concealed (so that assignments could not be predicted)?	Yes	NR	Yes	NR	Yes	Yes	NR	NR	NR
4. Were the study participants and providers blinded to treatment group assignment?	NR	NR	No	NR	NR	No	NR	NR	NR
5. Were the people assessing the outcomes blinded to the participants’ group assignments?	Yes	NR	Yes	NR	Yes	NR	NR	NR	NR
6. Were the groups similar at baseline on important characteristics that could affect outcomes (e.g., demographics, risk factors, co-morbid conditions)?	Yes	Yes	Yes	Yes	Yes	Yes	NR	NR	NR
7. Was the overall drop-out rate from the study at the endpoint 20% or lower of the number allocated to treatment?	No	Yes	No	Yes	Yes	Yes	Yes	Yes	Yes
8. Was the differential drop-out rate (between treatment groups) at endpoint 15 percentage points or lower?	Yes	Yes	Yes	Yes	Yes	Yes	Yes	Yes	Yes
9. Was there high adherence to the intervention protocols for each treatment group?	Yes	NR	Yes	NR	NR	NR	NR	NR	NR
10. Were other interventions avoided or similar in the groups (e.g., similar background treatment)?	Yes	No	Yes	Yes	Yes	NR	NR	NR	NR
11. Were outcomes assessed using valid and reliable measures, implemented consistently across all study participants?	Yes	No	Yes	Yes	Yes	No	No	No	No
12. Did the authors report that the sample size was sufficiently large to be able to detect a difference in the main outcome between groups with at least 80% power?	No	Yes	Yes	No	No	NR	Yes	No	No
13. Were outcomes reported or subgroups analyzed pre-specified (i.e., identified before analyses were conducted)?	Yes	Yes	Yes	Yes	Yes	Yes	Yes	NR	NR
14. Were all randomized participants analyzed in the group to which they were originally assigned, i.e., did they use an intention-to-treat analysis?	No	No	Yes	Yes	Yes	Yes	Yes	No	No
Quality rating (good, fair, or poor)	Poor	Poor	Fair	Fair	Fair	Poor	Poor	Poor	Poor
Additional Comments (If POOR, please state why):	the drop-out rates in both studies groups were over 40%, and no ITT analysis was done	participants who were using other services or support for smoking cessation were not excluded from the study	the dropout rates in both study groups were over 20%			all the data analysed were reported by the smokers’ spouse (i.e., non-smoking mothers)	group difference in demographic characteristics and smoking-related characteristics at baseline were not reported in this study	some important details about methodology were not reported in this study, and some participants in both groups were non-smokers	some important details about methodology were not reported in this study, the only assessment tool used in this study was developed by the authors without reliability and validity test

CD = cannot determine; NA = not applicable; NR = not reported; ITT = intent-to-treat.

**Table 2 ijerph-16-00107-t002:** Quality Assessments of Studies Included in the Systematic Review (pre-post studies).

	Huang et al., 2016 [31]	Liu et al., 2007 [26]	Meng et al., 2004 [23]	Huang, 2008 [27]
1. Was the study question or objective clearly stated?	Yes	Yes	Yes	Yes
2. Were eligibility/selection criteria for the study population pre-specified and clearly described?	Yes	No	No	Yes
3. Were the participants in the study representative of those who would be eligible for the test/service/intervention in the general or clinical population of interest?	Yes	CD	No	Yes
4. Were all eligible participants the met the pre-specified entry criteria enrolled?	Yes	CD	CD	Yes
5. Was the sample size sufficiently large to provide confidence in the findings?	NR	NR	NR	NR
6. Was the test/service/intervention clearly described and delivered consistently across the study population?	Yes	Yes	Yes	Yes
7. Were the outcome measures pre-specified, clearly defined, valid, reliable, and assessed consistently across all study participants?	Yes	Yes	No	Yes
8. Were the people assessing the outcomes blinded to the participants’ exposure/ interventions?	Yes	NR	NR	NR
9. Was the loss to follow-up after baseline 20% or less? Were those lost to follow-up accounted for in the analysis?	No	No	No	Yes
10. Did the statistical methods examine changes in outcome measures from before to after the intervention? Were statistical tests done that provided *p* values for the pre-to-post changes?	Yes	No	No	Yes
11. Were outcome measures of interest taken multiple times before the intervention and multiple times after the intervention? (i.e., did they use an interrupted time-series design)	No	No	No	No
12. If the intervention was conducted at a group level (e.g., a whole hospital, a community, etc.) did the statistical analysis take into account the use of individual-level data to determine effects at the group levels?	NA	NA	NA	NA
Quality rating (good, fair, or poor)	Fair	Poor	Poor	Fair
Additional Comments (If POOR, please state why):		some important details about methodology were not reported in this study, which makes the results unreliable	some important details about methodology were not reported in this study, which makes the results unreliable	

CD = cannot determine; NA = not applicable; NR = not reported.

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
