# Peer review of "Effectiveness of Interventions to Reduce Exposure to Parental Secondhand Smoke at Home among Children in China: A Systematic Review"

_ijerph, 2019, doi:10.3390/ijerph16010107_

Round 1

Reviewer 1 Report

The article is a review of Sino- and English-language articles on the SHS at home for children living with their parents in China. The selection of articles for the review has been described in detail (here the Authors got a little lost, because at the beginning they wrote that "Studies have to be published in the past 20 years"; were excluded if they were: 1) not published in Polish, or 2) not available in full text ". The description of the results begins with summary in how many publications (out of 13 selected to rag) and what was written about the problem. Then it is much better and all aspects of the problem have been deaply discussed. The conclusions summarize the analyzed research indicating what is missing in them and what it would have to be dealt with in the future.

Author Response

Point 1: The Authors got a little lost in the selection of articles for the review, because at the beginning they wrote that "Studies have to be published in the past 20 years"; were excluded if they were: 1) not published in Polish, or 2) not available in full text ".

Response: We would like to seek further clarification from the reviewer regarding this point. We stated our literature search aimed to identify studies published between January 1997 to December 2017 (p.3, lines 87-88) and that studies were excluded if they were 1) not published in English or Chinese, or 2) not available in full text (p.3, lines 112-113). 

Reviewer 2 Report

Review:

This is an interesting article reporting a systematic review of strategies used in intervention aiming at smoking cessation in China. The authors demonstrate that non-pharmacological approaches (eg. counseling, persuasion) could be helpful to successful quitting measured with different with end-points: lower cotinine levels in children, reduced tobacco consumption and self-reported increased quit rates. The introduction is understandable, well written.  The chapters on cultural differences between Western Societies and China are very interesting for an international reader (lines: 404-415, 420-424) and deserve to be developed.

The study was conducted in accordance with PRISMA Statement, and its methodology is extensively and clearly described. It does not require any revision nor supplementation.

Major comments:

The authors cite several previous systematic      reviews and meta-analyses on the same or similar issue published elsewhere      (Ref 14,15,16,17,18 and 19). However they did not discuss the      discrepancies and similarities between these papers and results obtained      by the authors. In respect to above-mentioned cultural differences.

Apart of quality assessment as presented in tables (Appendix A) this study deserves and graphical presentation of the extracted studies in the form of Table of Evidence to make it      easier for reader.

The discussion is long, however, the authors      only shortly mentioned limitation(s) of this study (lines 446-447). Please      address it. A good standard of writing a discussion would help a lot: 1.      principal findings;  2. Methodology;  3. Strong points and limitations of      the study; 4. Potential confounders; 5. Comparison with other studies;  6. Conclusions and future      research

 What really      disturbs me in the discussion was an engagement of the authors in reduction      of SHS  exposure and solutions      provided by them in this context. Although I do understand importance of the problem, I’m really not convinced enough to propose specific solutions      that are highly speculative within a scientific article and it does not      contribute much to general understanding of the results obtained (eg. lines      370-374, 393-395, 424-427). The importance of the author’s findings became      also lost..

The real novelty of the findings should be      emphasized both in the abstract and in the discussion.

Minor comments:

Line 20-21. The sentence “The results were lower cotinine levels in children (n=2), and reduced tobacco consumption (n=5) and increased quit rates (n=6) among parents” is unclear. Rephrasing suggested.

The chapters on cultural      differences between Western Societies and China are very interesting for      an international reader (lines: 404-415, 420-424) and deserve to be      developed.

Author Response

Point 1: The authors cite several previous systematic reviews and meta-analyses on the same or similar issue published elsewhere (Ref 14, 15,16,17,18 and 19). However they did not discuss the discrepancies and similarities between these papers and results obtained by the authors, in respect to above-mentioned cultural differences.

Response: we have added a comparison between results from previous systematic reviews and meta-analyses and our study in the discussion part.

Point 2: Apart of quality assessment as presented in tables (Appendix), this study deserves graphical presentation of the extracted studies in the form of Table of Evidence to make it easier for reader.

Response: We have inserted the table of characteristic of included studies in the main text, but we had some difficulties in inserting the table because of the big size of the table. We will communicate with the editor about this problem.

Point 3: The discussion is long, however, the authors only shortly mentioned limitation(s) of this study (lines 446-447). Please address it. A good standard of writing a discussion would help a lot: 1. principal findings; 2. Methodology; 3. Strong points and limitations of the study; 4. Potential confounders; 5. Comparison with other studies; 6. Conclusions and future research

Response: we have made a revision in the limitation section and have rearranged the organization of the discussion according to your suggestion.

Point 4: What really disturbs me in the discussion was an engagement of the authors in reduction of SHS exposure and solutions provided by them in this context. Although I do understand importance of the problem, I’m really not convinced enough to propose specific solutions that are highly speculative within a scientific article and it does not contribute much to general understanding of the results obtained (eg. lines 370-374, 393-395, 424-427). The importance of the author’s findings became also lost. The real novelty of the findings should be emphasized both in the abstract and in the discussion.

Response: We have proposed the possible solutions based on present findings, and find it important to extend our discussions based on the cultural/ socioeconomic context and other relevant research on tobacco smoking previously conducted in China. However, have further reframed those possible solutions as suggestions for future practice/ research investigation throughout.

Point 5: Line 20-21. The sentence “The results were lower cotinine levels in children (n=2), and reduced tobacco consumption (n=5) and increased quit rates (n=6) among parents” is unclear. Rephrasing suggested.

Response: we have changed the sentence to “Several encouraging results were observed, including lower cotinine levels in children (n=2), reduced tobacco consumption (n=5), and increased quit rates (n=6) among parents.”

Reviewer 3 Report

This is a well written and comprehensive systematic review examining the impact of smoking cessation programs on Chinese children's exposure to second hand tobacco smoke in the their homes (in China, Hong Kong, Macau and Taiwan). The reporting of their methodology and findings are clear, their discussion links to their findings and they make some useful recommendations for future research considerations. There are few minor typographical points that could be improved: 1. On page 9, line 333 - i think the term 'urban' could be used instead of 'developed'; 2. Page 10, line 392 - 'areis' could be replaced with 'can be'. 3. Check the formatting of their references for consistency in style.

Author Response

Point 1: on page 9, line 333. I think the term “urban” could be used instead of “developed”.

Response: The term “developed” has been revised to “urban” . 

Point 2: page 10, line 392-“areis” could be replaced with “can be”.

Response: thank you very much for your suggestion, we have amended the typo.

Point 3: check the formatting of their references for consistency in style.

Response: we have made amendments to the reference list.